# Analogical reasoning in first and second languages

**Miki Ikuta** [1] *, **Koji Miwa** [2]

**1** Waseda University, Shinjuku-ku, Tokyo, Japan, **2** Graduate School of Humanities, Nagoya University, Nagoya, Aichi, Japan

* ikutam@aoni.waseda.jp

**Data Availability Statement:** The full set of stimuli is not in the data set due to copyright concern. The data underlying the results presented in the study are available on the Open Science Framework (https://osf.io/bzegd/).

## Abstract

This study investigated how linguistic predictors such as word frequencies, the difficulty and creativity of problems, and the category of problems contribute to analogical reasoning in L1 and L2. This study also investigated how different types of similarities (i.e., perceptual and relational similarities) are processed in analogical reasoning. In Experiment 1, Japanese participants were asked to solve 100 multiple-choice A:B::C:D analogy problems (e.g., skeleton: bone:: tornado: wind) in their first language, Japanese (L1). In this experiment, participants also rated the difficulty and creativity of problems. In Experiment 2, Japanese participants completed the same tasks, but the problems were shown in their second language, English (L2). The results showed that problems presented in L1 elicited higher accuracies and faster response times than in L2. A significant interaction was found between languages (L1/L2) and the category of problems which indicates that finding a perceptual similarity (e.g., the shape image of word concepts) with verbal stimuli in L2 is more challenging than in L1. Moreover, our results on response times indicated that processing relations between words would be carried out in L1 without any specific instruction while it would not be completed in L2 possibly due to the cognitive demand related to lexical processing. Considering these results, it is advisable in an educational setting to provide L2 learners with enough time and explicit instruction on understanding word relationships when forming analogies.

## Introduction

### Analogical reasoning

Retrieving prior knowledge and transferring it to the new domain is a fundamental cognitive process when people learn something new [1]. This process is called analogical reasoning, and numerous studies have been conducted in the related areas of cognitive psychology, education and artificial intelligence [2]. In everyday life, adults use verbal analogies for efficient learning. For example, cell metabolism can be explained with a metaphor, "Mitochondria are the furnace (or powerhouse) of the cell" [3, p. 187], knowing that mitochondria need glucose to generate energy, and a furnace needs fuel to burn fire. Thus, the verbal analogy enhances the

**Funding:** This research has been supported by Kaken Grant-in-Aid for Japan Society of Promotion of Science Fellows (21J15484). The funders had no role in study design, data collection and analysis, decision to publish, or preparation of the manuscript.

**Competing interests:** The authors have declared that no competing interests exist.

understanding of the concept of mitochondria by highlighting the similar relation between mitochondria/oxygen and furnace/fuel. As such, analogical reasoning contributes to human language-oriented performances in a wide range of situations, such as classroom learning/teaching [4, 5], scientific reasoning [6–8] and foreign language metaphor processing [9].

Studying analogical reasoning in L2 in today's global society is important because the vast majority of people use two or more languages. However, very few studies have examined analogical reasoning in bilingual participants. While research has shown that analogical reasoning enhances language learning in certain educational settings [10], its underlying mechanism remains underexplored in second language acquisition. The present study investigates both accuracy and reaction times in analogical problem solving in L2, while also shedding light on the role of similarity and linguistic predictors, such as word frequency. The following section presents findings on analogical reasoning in L2, followed by a discussion on their relation to different types of similarity (i.e., perceptual and relational).

## Analogical reasoning in L2

One of the earliest studies on bilinguals' analogical reasoning is Francis [11]. She investigated whether knowledge acquired in one language could be transferred to the situation of problem solving in another language. In her experiment, she found that participants were able to solve the target problem in either L1 or L2 after being presented with the source problem in a different language. The findings were replicated by Fukumine and Kennison [12] indicating that language consistency had no significant effect on analogical transfer between the source and target problems. Thus, as summarized in [13], these two studies revealed that knowledge acquired in either L1 or L2 is transferable in analogical reasoning.

Even though Francis tentatively concluded that analogical transfer while reading two structurally similar stories is language-free, she also pointed out that the process of analogical reasoning is sensitive to language proficiency level, especially because of the cognitive load caused by unfamiliar languages. The negative foreign language effect was investigated in Wakebe et al. [14]. In their study, Japanese learners of English first read a source problem in English (L2). Then, they were assigned to solve the target analogical problem either in Japanese (L1) or English (L2). The study reported that participants performed better in the L1 condition than the L2 condition. Taken together, solving target problems in a less familiar/proficient language through analogical transfer is possible, but more difficult than doing the same in L1.

These prior studies focused on the accuracy of problem solving in L2, rather than investigating the underlying mechanisms of the reasoning process itself. In analogical reasoning, knowledge transfer between domains occurs based on similarities [3]. However, how this similarity information is transferred in L2 remains unexplored. In addition, simply investigating the accuracy of analogical reasoning is not sufficient to fully understand at what point the differences between L1 and L2 reasoning processes occur.

## Similarities in analogical reasoning

Aside from the mechanism of L2 analogical reasoning, the effects of different types of similarity in verbal analogical reasoning remain insufficiently explored, too. In analogical reasoning, transfer of the information between domains is crucial. The information is usually based on a certain type of similarity shared between domains [3]. According to Behrens [15], there are two similarities such as perceptual and relational similarities in analogical reasoning. The perceptual similarity is a perceivable similarity such as colors and shapes. On the other hand, the relational similarity is an unperceivable similarity that two domains share through meaning associations such as roles and functions [16]. Psychological studies have used image stimuli to

study the influence of perceptual similarity even though instruction was given to find relational similarity while solving analogical reasoning [17–20]. For example, in Christie and Gentner [19], to match with a source picture of a black dog located above a white dog, children preferred the shared objects (e.g., a picture of a black cat and dog in a random position) over shared relational patterns (e.g., positional relations with colors: a black bird is located above a white bird). In addition, perceptual similarity facilitates adults' analogical reasoning indicating perceptual information is crucial for both adults and children's reasoning processing as summarized in Kucwaj et al [21]. However, the role of perceptual similarity is relatively less studied in mentally healthy adults [21]. To investigate the mechanism of how the mental imagery through lexical processing would affect reasoning processes, further research is needed. Moreover, as mentioned, image stimuli have been often used in psychological studies to investigate perceptual information in analogical reasoning. Therefore, how objects' perceptual image within word's concepts is processed in verbal analogical reasoning is unknown.

## Visual and linguistic information during lexical processing

As mentioned above, the different types of similarity have received limited attention in verbal analogical reasoning research, but the processing mechanism of visual and semantic information within lexical processing has been widely studied in monolingual and bilingual research [22–25]. For example, Huettig and Hartsuiker [23] investigated how eye movements would be affected by visual and semantic competitors during naming tasks. During the experiment, participants heard a verbal input. Then, they chose an appropriate picture and named it. The most important finding in their study was that participants paid attention to competitors such as visually similar objects (comparing a ladle to a saxophone as they share the same long shape) and semantically related objects (comparing a stove to a plate as they share the same category membership) but not unrelated objects. The study reported that both visual imagery and semantic information can be activated by verbal input. A general framework introduced by Huettig et al. [26] explains that perceptual and semantic information can be triggered differently depending on the input type. According to their framework, although verbal and visual inputs can activate both semantic and visual representations stored in one's long-term memory, how activation is achieved would be different depending on the modality. For example, they describe verbal (spoken) linguistic input that activates phonological and semantic structures first, followed by their associated visual representations. In contrast, visual input such as pictures activates visual representations first and later activates semantic and phonological information if sufficient time is given [see also, 25].

The mechanism of visual and semantic information with verbal inputs has also been investigated in bilingual research. One noteworthy study exploring the effects of perceptual similarity (e.g., the shape image of word concepts) and the influence of a foreign language is Hayakawa and Keysar [22]. The primary focus of their study was to investigate how using a foreign language reduces mental imagery. In their experiments, bilingual participants performed worse in L2 than in L1 when choosing the most different shape or category among three words (e.g., carrot, mushroom, and pen, where carrot has a similar shape to pen, whereas carrot and mushroom are in the same category). More importantly, the accuracy of judging a shape among words in L2 was significantly lower than that of judging category membership. The results support their hypothesis that the use of a foreign language would lead to a reduction in mental imagery, although it was less effective when exploring the category membership of words than when exploring the shape of the conceptual image of the words.

Although visual information processing with lexical input may offer valuable insights into the area of verbal analogical reasoning in L1 and L2, these two lines of research have been studied separately. Integrating knowledge from these different topics helps investigate how perceptual information is processed during verbal analogical reasoning.

## Research questions in the current study

The present study offers new insights into both bilingual and psychological studies by investigating the mechanism of problem solving in L1 and L2. Specifically, the present study highlights the types of similarity and language conditions through the three research questions below.

1. Does the language difference (L1/L2) influence the reaction times (RTs) and accuracy of analogical reasoning?
   The previous studies suggested that the use of a foreign (unfamiliar) language would result in lower accuracy for analogical reasoning [10, 14]. As in the previous study, we hypothesized that accuracy would be lower in the L2 condition. This is because less efficient language processing in L2 requires great cognitive load, leading to limited cognitive resources to handle analogical problems. In addition to accuracy, this study analyzed three types of reaction times: the time to read analogies (e.g., delight: happiness:: woman: ???), the time to choose an answer (e.g., lady) from multiple choices (e.g., lady, man, success and TV), and the combined time for both. Like accuracy, we hypothesized that the greater cognitive load would result in longer response time in the L2 condition. The reaction times were analyzed to investigate at what point processing differences occur in analogical reasoning in a less familiar language (L2).

2. To what extent do perceptual and relational similarities influence the reaction times and accuracy of verbal analogical reasoning?
   Considering that verbal input typically activates semantic information before mental imagery as suggested by Huettig et al. [26], we hypothesized that the problems based on perceptual similarity of the word concept would require longer response times compared to those based on relational similarity in this study using verbal analogies. In terms of accuracy, solving problems based on perceptual similarity may not differ significantly from that of relational similarity in terms of accuracy in L1 condition, but in L2 condition. This is because as suggested by Hayakawa and Keysar [22], the use of a foreign language reduces mental imagery of words, although this effect is less pronounced when exploring category membership. To test these hypotheses, separate analyses of accuracy and reaction times were conducted in each language condition.

3. Is there a significant interaction between the similarities and languages?
   Assuming that using less proficient languages is cognitively demanding, solving analogies in L2 would cause lower accuracy and longer reaction times compared to L1 for all categories. However, it is also possible that certain categories can be solved more easily in both L1 and L2 conditions. For instance, Jones et al. [27] reported higher accuracies for categorical analogies (e.g., hammer: tool) compared to compositional and causal analogies (e.g., allergy: sneeze). Therefore, assuming that the higher-level conceptual representations are comparable between L1 and L2 participants, it can be hypothesized that the accuracy for synonym analogies would be higher in both L1 and L2 conditions, while the accuracy between languages would vary for other categories. This hypothesis was tested by directly comparing the L1 and L2 conditions, with a specific focus on examining the interaction between problem categories and languages.

## Methodological considerations

**Word frequency.** Although previous studies often referenced cognitive load to explain the inefficiency of L2 reasoning, its precise nature has been rarely clarified. Word frequency is one of the most influential linguistic predictors affecting bilinguals' reading behavior. In general, the higher the word frequency, the faster people process the words [28–32]. This is because the more frequently people encounter a word, the more easily they can access its lexical information. Assuming people are less exposed to their L2 than their L1, it is expected that the effects of word frequency are greater in L2 than in L1, as found in previous studies [28–32]. Additionally, low-frequency words are expected to hinder efficient analogical transfer to a greater extent, especially when solving problems. For instance, *filament* is a relatively low-frequency word that appeared in a story used in [14]. This can influence the speed of processing at any stage of analogical reasoning in L2, but it may not be the case in L1. Previous studies using stories as materials did not consider the word frequency that might be crucial for analogical processing research. Considering word frequency as a linguistic predictor, this study investigated the foreign language inhibitory effect on not only the accuracy but also the processing speed of analogical reasoning.

**Difficulty and creativity of analogy.** In addition to the word frequency effect, recent studies have noted the importance of considering predictors such as difficulty and creativity [2, 27, 33–35]. Difficulty and creativity are similar concepts but have some differences. On the one hand, the difficulty of problems refers to the complexity in the analogical transfer, which contrasts with the unfamiliarity of words (e.g., "How difficult is it to generate a solution for this analogy?," Green et al. [34], p. 266). On the other hand, creativity refers to the semantic distance between relations of the first domain (A:B) and the second domain (C:D). For example, the semantic distance between kneepad: knee and helmet: head is closer than that between atmosphere: earth and helmet: head, although they both share the relations of location [2]. Both the difficulty and creativity of problems were assumed to show significant effects on accuracy and response times; the more difficult or creative the problems are the less accurate participants' responses would be, and the longer response times participants would require. To the best of our knowledge, a database of semantic distance between the relations of words measured by bilingual participants does not exist. Collecting human-rated measures is needed to better understand the effect of these measures on L1 and L2 analogical reasoning. To overcome this issue, this study opted for subjective rating tasks with a simpler form called the four-term A:B::C:D verbal analogy. This item type allowed us to collect decent amount of data (for 100 problem sets) and consequently allowed us to test the foreign language effects, with frequency of words and difficulty and creativity of the problems all included as predictors simultaneously in regression models.

## Experiment 1

In Experiment 1, the judgment task on analogical problems was conducted in Japanese. In this task, participants solved 100 analogical reasoning problems in their first language (Japanese). In addition, participants also rated the difficulty and creativity of problems during the experiment.

## Method

**Participants.** The participants were 44 Japanese university students, but 2 of them were excluded from the data analysis due to a technical error during data collection. The remaining 42 participants' (20 males/19 females/3 no responses) ages varied from 19 to 29 ($M = 21.36$, $SD = 2.47$). Their average English vocabulary size tested through the Vocabulary Size Test [36]

was 6811.9 (*SD* = 1255.47). For all participants, their self-rated L1 proficiencies (listening, reading, speaking) were higher than their corresponding L2 proficiencies according to the Language Experience and Proficiency Questionnaire (LEAP-Q) [37]. None of them reported that they had spent more than one year abroad or had any learning disabilities. All participants received a 1500-yen gift card for their participation. The experiment was carried out in accordance with the ethics protocols approved by the ethics committee of Graduate School of Humanities, Nagoya University (Ref No. NUHM-21-008). The data collection started on April 19th, 2022, and ended on July 8th, 2022. Participants were provided with written informed consent in their L1 (Japanese) through google form. To participate in this study, they typed their names and the dates of their agreement.

**Materials.**   To prepare the stimuli, we referred to relations and problem sets in Dahan and Tanenhaus [38], De Groot et al. [39], Gladkova et al. [40], Lu et al. [41], Popov et al. [42], and Yee et al. [43], as well as educational materials [44–46]. However, some words in the previous studies were low-frequency words. For example, according to Gimenes and New's [47] database of Twitter, blog, and newspaper frequencies, cantaloupe (frequency = 4.02) was a very low-frequency word. We assumed it would be unknown to Japanese learners of English. Therefore, we replaced the word with *melon* (frequency = 8.94), which was relatively more frequent. In addition, instead of employing the original problems in the previous studies as they were, we carefully matched the relations to construct our problems with cross-domain relations. Within-domain and cross-domain are predictors that influence the speed and accuracy of analogical reasoning. For example, "blindness: sight:: deafness: hearing is a relatively near analogy, whereas blindness: sight:: poverty: money is more distant" [2, pp.1806-1807]. The retrieval of an analogy from the same domain is usually easier than the retrieval of an analogy from a different domain. Thus, we decided to sort the problems to avoid any possible effects derived from the problem feature (within- or cross-domain). In addition to the items used in the past studies, we prepared items and ended up with 100 problems in total with an equal number in each category: 20 object matches (OBJ), 20 synonyms (SYN), 20 antonyms (ANT), 20 location-time-context (LTC) items, and 20 material-part-whole (MPW) items. Examples are shown in Table 1.

All analogical reasoning problems were shown in the form of A:B::C:D (e.g., delight: happiness:: woman: lady). The problems appeared on the screen with the last word replaced by??? (e.g., delight: happiness:: woman: ???). The participants were presented with four choices and were asked to choose the correct answer for D to complete A:B::C:D. Multiple choices consisted of D (e.g., lady) as the answer and related words of the word C (C', e.g., man), B (B', e.g., success), and an anomalous word (Anom, e.g., TV) as competitors. Related words of C and B were used to measure how well participants suppressed the influence of B and C words during analogical reasoning. Although suppression of irrelevant words was not a focus of the current study, we decided to prepare those options to control the quality of the choice-making process, as Jones et al. [27] suggest that the semantic association between words in problems and

**Table 1. List of items.**

| Category | Sub-Category | # of Items | Examples (L1:Japanese) | Examples (L2: English) |
|---|---|---|---|---|
| Visual | Object (OBJ) | 20 | ビスケット:レンズ:: 辞書:パスポート | biscuit:lens:: dictionary:passport |
| Relational | Antonym (ANT) | 20 | ヒーロー:腰抜け:: 製品:模造品 | hero:coward:: product:knockoff |
| | Synonym (SYN) | 20 | 初心者:新米:: 癖:慣習 | beginner:novice:: habit:tradition |
| | Location, Time, Context (LTC) | 20 | 騎士:城:: ランナー:走路 | knight:castle:: runner:track |
| | Material, Part, Whole (MPW) | 20 | 骸骨:骨:: 竜巻:風 | skeleton:bone:: tornado:wind |

response choices influences accuracy. To overcome this issue, we conducted a word association test to sample related words of A, B, C, and D. Details of the word association test are available in the S1 Appendix.

**Procedure.**  All the tasks, including two rating tasks (difficulty and creativity) and the LEAP-Q, were conducted online. Participants were asked to find a quiet place with a stable internet connection and to use a PC with a mouse. Rating tasks were programmed with PsychoPy2 [48] and hosted on the Pavlovia platform at pavlovia.org. In both rating tasks, the background color was dark gray, and the letters were white, except that a correct word was presented in red after participants responded to a reasoning problem. Two rating tasks were conducted separately. In both tasks, before they began the main sessions, participants completed practice sessions. The practice session consisted of one problem from each category of relations, such as OBJ, SYN, ANT, LTC, and MPW. Participants were able to practice as many times as they wanted. Instructions in the practice trials were all prepared in Japanese (L1). The order of 100 problems and the positions of multiple choices were randomized. During the experiment, their rating scores, as well as the accuracy of their responses and the time spent reading problems and choosing answers, were measured.

The sequential procedure is illustrated in Fig 1. First, participants were asked to join a practice session for the difficulty rating task. After the practice session, they were asked to complete the difficulty rating task. Then, participants were guided to the creativity rating task starting with a practice session and followed by a main session. In the difficulty rating task, participants first saw an A:B pair (e.g., delight: happiness) on the upper side of a screen and were asked to click a ClickHere button. After they clicked the button, an additional C:??? (e.g., woman: ???) pair appeared with the A:B pair remaining on the screen. When they clicked on the ClickHere button again, a box with four choices appeared at the center of the screen, with questions A:B and C:??? remaining on the screen. After they chose an appropriate word, the correct answer became highlighted in red. At the same time, a seven-point scale appeared at the bottom of the screen with numbers from one to seven. At this point, participants were asked to rate how difficult the problem was (1-easy, 7- difficult).

After the difficulty rating task, participants were guided to the creativity rating task. In this task, participants first saw an A:B pair (e.g., delight: happiness) on the upper side of a screen and were asked to click on the button ClickHere. After they clicked on the button, an additional line of the C:D pair (woman: lady) appeared, with the A:B pair remaining on the screen. Different from the difficulty rating task, the word D (e.g., lady) appeared on the screen. Then, participants were asked to rate how surprising the C:D pair was, given the A:B pair.

## Results and discussion

**Accuracy.**  The data set is available on the Open Science Framework (https://osf.io/bzegd/). The full set of stimuli is not in the data set due to copyright concern, but available by contacting the first author. The data of 42 participants were analyzed. All 42 participants in Experiment 1 responded with over 70% accuracy. Responses longer than 30,000 ms were omitted as outliers. Overall, 98% of the data were kept for further analyses. Although both difficulty and creativity were rated in Experiment 1 with separate tasks, they were highly correlated ($r = .95, p < .001$). Thus, difficulty and creativity were analyzed in separate models to avoid multicollinearity. Accuracy and reaction times were analyzed through (generalized) linear mixed-effects modeling [49, 50].

As fixed effects we included the category of problems (levels: OBJ, ANT, LTC, MPW, and SYN) and rating scores coding difficulty (range: 1–7). In addition, it was expected that the participants might have felt more tired or easier as they experienced the same tasks over and over

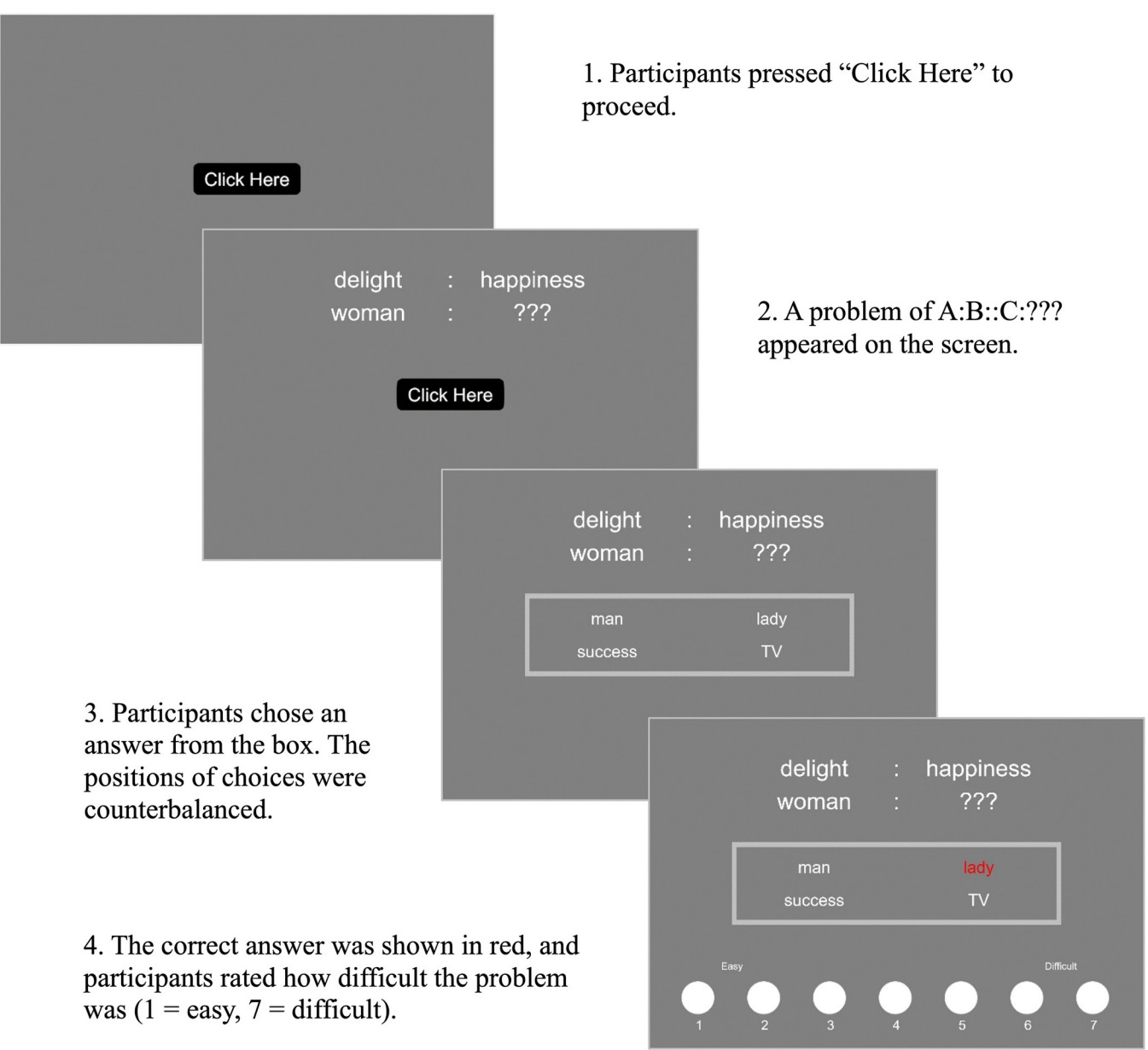

**Fig 1. Procedure of the difficulty rating task.**

again. To statistically control for the possible fatigue and adaptation effect in experiments, previous psycholinguistic studies included trial count as a control variable in regressions models [e.g., 9, 32, 51–53]. Therefore, trial count (i.e, the number of preceding trials, ranging from 1 to 100) was also included in our analysis to consider a possible change of participants' response patterns throughout the 100 trials. The numerical predictors were standardized.

To study the frequency effects, Twitter, blog, and newspaper frequencies per million [47] were used for two reasons. First, because people experience analogical reasoning in a wide range of situations, the frequency scores obtained from long and short texts as well as formal and casual contexts were appropriate in this research. Second, Twitter, blog, and newspaper frequencies have lists of frequency in 66 languages, including Japanese and English. It made it

possible to analyze both English and Japanese items fairly with the frequency of words obtained from the same sources. To analyze the results of Experiment 1, we used Japanese word frequency.

We opted for forward-fitting. The first model included the category of problems, standardized trial count, and standardized rating score of difficulty as fixed effects, together with random intercepts for participants and items. Then, from the second models on, we tested the log-transformed frequencies of words A, B, C, D, B', C', and Anom one by one in the order that participants encountered them. That is, the frequency of word A in A:B::C:??? /D B' C' Anom was added to the model first, and then the frequency of word B followed. We kept a given frequency predictor if it was significant and removed it from the model if it was not.

The final model contained category, trial count, and difficulty as fixed effects, random intercepts for participants and items, and by participant random slopes of trial count and difficulty. In this final model, difficulty was significant ($p < .001$). The category of problems was not significant. No word frequency measure was significant. A pairwise comparison with Tukey's adjustment revealed that there was no significant difference between categories.

We also fitted another model replacing difficulty ratings with creativity ratings, with all other fixed and random effects remaining unchanged. The effect of creativity was significant ($p < .001$). Trial count was not significant. Unlike the model with difficulty, the accuracy of OBJ was significantly lower than that of ANT ($p < .001$), and the accuracy of ANT was significantly higher than that of LTC ($p = .017$).

In the analysis of the accuracy with difficulty, there was no significant difference between the five categories (OBJ, ANT, LTC, MPW, and SYN), whereas there were some significant differences in the model with creativity. Nevertheless, the model with difficulty had a larger R-squared than the model with creativity ($R^2 = .76$ and $R^2 = .54$, respectively). Thus, difficulty would be a better predictor that explains the larger amount of variance in the data.

**Total Time (Reading Time + Solving Time).** Next, three different reaction times of correct responses were analyzed. First, we analyzed the Total Time, which is the time that participants spent reading problems A:B::C:??? (Reading Time) and solving problems by clicking on an appropriate word (Solving Time). In addition, Reading Time and Solving Time were analyzed separately to investigate how much time participants needed to learn the relation of the first domain and to solve the problem by transferring the knowledge from the first domain to the new domain. Taking this approach, we can also investigate the time course of potential language effects; Reading Time and Solving Time may be codetermined by different variables for L1 and L2. To analyze reaction times, linear mixed effect modeling was used. Reaction times were all log-transformed. Similar to the accuracy analysis, the model fitting started with fixed effects such as category, difficulty and trial count and random intercepts for participants and items. Log-transformed frequencies were analyzed one by one starting with word A. Similar to the analysis of accuracy, word frequency did not codetermine Total Time significantly. Random slopes were included to improve the model. In the final model, category, trial count, and difficulty were included as fixed effects. The model also contained random intercepts for participants and items, as well as by participant random slopes for trial count and difficulty. In this final model, difficulty and the trial count was significant ($ps < .001$). A pairwise comparison between categories revealed that reaction time for OBJ was significantly longer than the one for other four categories were significantly different ($ps < .001$). In addition, the reaction time for the ANT was significantly shorter than that for the LTC ($p = .006$).

A model with creativity was also fitted with the same fixed and random effects. The effects of creativity and trial count were significant ($ps < .001$). Similar to the model with difficulty, there were significant differences between OBJ and other categories ($ps < .001$); the total time

of OBJ was significantly longer than that of other categories. Additionally, the total time of ANT was significantly shorter than that of LTC ($p = .004$).

**Reading Time.**   The reading time was analyzed in the same manner. The final model contained category, trial count, and difficulty as fixed effects, as well as random intercepts for participants and items and by participant random slopes for trial count and difficulty. In this final model, difficulty was significant ($p < .001$), and the trial count was significant ($p < .001$). A pairwise comparison between categories revealed that differences in Reading Time between relations of OBJ and ANT ($p < .001$), OBJ and MPW ($p = .0156$), and OBJ and SYN ($p = .0132$) were significant; OBJ elicited longer response times than ANT, MPW and SYN. ANT elicited a shorter response time than LTC ($p = .0276$).

A model with creativity was also made with the same fixed and random effects. The effects of creativity ($p = .0385$) and trial count ($p < .001$) were significant. Similar to the model with difficulty, there were significant differences between OBJ and some categories; the Reading Time of OBJ was significantly longer than that of ANT ($p < .001$), MPW ($p = .008$) and SYN ($p = .007$). Additionally, the Reading Time of ANT was significantly shorter than that of LTC ($p = .0167$).

**Solving Time.**   The Solving Time was analyzed in the same manner. The final model included the category, trial count, and difficulty as fixed effects, as well as random intercepts for participants and items and by participant random slopes for trial count and difficulty. In this final model, trial count ($p = .0131$) and difficulty ($p < .001$) were significant. A pairwise comparison between categories revealed that the reaction time of OBJ was significantly slower than that of other categories ($ps < .001$).

A model with creativity was also fitted with the same fixed and random effects. The effects of creativity and trial count ($ps < .001$) were significant. Similar to the model with difficulty, there were significant differences between OBJ and all categories; the Solving Time of OBJ was significantly longer than that of the other categories ($ps < .001$). Model summaries are available in S2 Appendix.

## Summary of Experiment 1

In Experiment 1, participants solved analogical reasoning problems in L1. In addition, they rated the difficulty and creativity of the problems by using a seven-point Likert scale (1-easy, 7- difficult). The models with difficulty had a larger R-squared than the model with creativity in all analyses of accuracy and reaction times. The category of problems was not significant in the analysis of accuracy with difficulty as a fixed effect. In addition, it is noteworthy that the separate analysis of the Reading Time and Solving Time enabled us to investigate the time course of analogical reasoning with different categories. The analysis of the Total Time revealed significant differences between categories; OBJ took longer to read and solve than the other categories, and LTC took longer to read and solve than ANT. The analysis of the Reading Time revealed that some of those significant differences had already emerged when reading A: B::C:???. In this early phase, participants had not yet encountered multiple choices. Thus, consciously or not, in the earliest time frame, participants already started juggling with the problem-related factors that varied from categories when learning the relation of the first domain. Overall, the results of Experiment 1 not only demonstrated the significant effect of difficulty and creativity as the previous study suggested but also revealed that the category of problems influences how people learn the first domain and how they transfer it to the next domain. The significant differences were only observed for reaction times but not accuracy. This is in line with the hypothesis; with verbal input, visual information requires longer time to be accessed because it is available only after semantic information, as Huettig et al. [26] suggests.

## Experiment 2

In Experiment 2, we conducted the same experiment but with items in English (L2) and with different participants. This was to investigate whether the pattern of the results observed in Experiment 1 would be observed similarly when solving problems in a foreign language.

### Method

**Participants.**   The participants were 47 Japanese university students. The data of six participants were discarded because their accuracy was lower than 50%. The remaining 41 participants' ages (21 males/20 females) varied from 18 to 30 ($M$ = 20.68, $SD$ = 2.58). Their scores on the Vocabulary Size Test [36] showed that they were between intermediate-level and upper-level learners of English ($M$ = 7217.07, $SD$ = 1475.45). A $t$-test revealed that the participants' vocabulary size was not significantly different between groups of participants in Experiment 1 and Experiment 2 ($t$(78.34) = -1.35, $p$ > .05).

For all participants, their rated L1 proficiencies (listening, reading, speaking) were higher than their corresponding L2 proficiencies according to LEAP-Q [37]. None of them reported that they had spent more than one year abroad or had any learning disability. All participants received a 1500-yen gift card for their participation. The data collection started on January 30th, 2022, and ended on March 3rd, 2022. Participants were provided with written informed consent in their L1 (Japanese) through google form. To participate in this study, they typed their names and the dates of their agreement.

**Material.**   Materials were the same as in Experiment 1, but all were shown in English (L2).

**Procedure.**   As in Experiment 1, all the tasks, including two rating tasks (difficulty and creativity), the Vocabulary Size Test, and LEAP-Q, were conducted online. Instructions in the practice trials were all prepared in Japanese (L1), but all the analogical problems in the two rating tasks were shown in English (L2).

### Results and discussion

**Accuracy.**   The data of 41 participants were analyzed after discarding 6 participants' data whose accuracy was lower than 50%. Responses longer than 30,000 ms were omitted as outliers. Overall, 98% of the data were kept for further analysis. First, accuracy was analyzed with generalized linear mixed-effects modeling. In the final model, the category of problems, standardized trial count, and standardized difficulty rating score were included as fixed effects. The model also contained random intercepts for participants and items, as well as by participant random slopes for trial count and difficulty. The results showed a significant effect of difficulty ($p$ < .001). The trial count was not significant ($p$ > .05) after including a by participant random slope for trial count. A pairwise comparison with Tukey's adjustment revealed that there was a significant difference between OBJ and SYN ($p$ < .001); OBJ elicited more incorrect responses than SYN.

A model with creativity instead of difficulty was also fitted with the same fixed and random effects. The results revealed a significant effect of creativity ($p$ < .001), whereas trial count was not significant ($p$ > .05). There were significant differences between categories: OBJ received more incorrect responses than ANT, MPW and SYN ($ps$ < .001), and LTC received more incorrect responses than SYN ($p$ = .0252).

The results of accuracy for Experiment 2 revealed new findings that were not seen in the results of Experiment 1. First, the accuracy of OBJ was significantly lower than that of SYN even after controlling for the effects of difficulty. This indicates that finding perceptual similarity in two words' concepts might be more challenging than finding relational similarity when a verbal input is in L2. In other words, the disadvantage of L2 use might be more explicit when

evoking perceptual information of words' concepts than when evoking relational information of words' concepts. This issue will be further discussed in the general discussion, recalling Hayakawa and Keysar's study [22].

**Total Time (Reading Time + Solving Time).** Similar to Experiment 1, reaction times of correct responses were separated into the Reading Time and Solving Time. First, the Total Time, which is the sum of the Reading Time and Solving Time, was analyzed with linear mixed-effects modeling. The final model included the category, trial count, difficulty and frequency of the first word as fixed effects, random intercepts for participants and items, and by participant random slopes for trial count and difficulty. Regarding the frequency, we used log-transformed word frequency from Twitter, blog, and newspaper frequencies for English [47]. In this model, difficulty, trial count and first word frequency were significant ($ps < .001$). A pairwise comparison with Tukey's adjustment between categories revealed that the reaction times between relations of OBJ and the other four categories such as ANT, LTC, MPW and SYN were significantly different ($p < .001$); OBJ took longer to be responded to compared to the other categories.

A model with creativity revealed significant effects of creativity ($p < .001$), first word frequency ($p < .001$) and trial count ($p = .001$). There were significant differences between categories: OBJ received a longer Total Time than ANT, LTC, MPW and SYN ($ps < .001$). Although the difference between ANT and LTC was not significant, the results of the Total Time in Experiment 2 were similar to those in Experiment 1, as the analysis revealed that OBJ had a significantly longer reaction time than the other categories. To further analyze the time course of analogical reasoning in L2, the Reading Time and Solving Time were analyzed separately.

**Reading Time.** The time that participants spent reading problems A:B::C:??? was analyzed as the Reading Time. The final model included the category, trial count, difficulty, and the log-transformed frequency of the first word and second word as fixed effects, random intercepts for participants and items, and by participant random slopes for trial count and difficulty. The results revealed significant contributions of the trial count ($p < .001$), frequency of the first ($p = .0061$) and second words ($p < .001$), and difficulty ($p = .0197$). There were no significant differences between categories ($ps > .05$).

The model with creativity revealed significant effects of the first word frequency ($p = .0045$), the second word frequency ($p = .0004$), and trial count ($p < .001$). The effect of creativity was not significant ($p > .05$). There were no significant differences between categories ($ps > .05$).

Surprisingly, there were no significant differences between categories in the analysis of the Reading Time in Experiment 2. The nonsignificant contribution of creativity further supports the explanation that participants read the problems but did not process the relations thoughtfully. Even so, the analysis of the Total Time revealed significant differences between categories. Although the categorical differences merged in both the Reading Time and Solving Time in Experiment 1, the differences might have merged in the Solving Time in Experiment 2. To test this assumption, the Solving Time was analyzed.

**Solving Time.** The duration between the time that the multiple choices were presented, and the time participants took to choose the answer was analyzed as the Solving Time. The final model included category, trial count, difficulty, and log-transformed frequency of the first word as fixed effects, random intercepts for participants and items, and by participant random slopes for trial count and difficulty. The results revealed that the difficulty ($p < .001$) and frequency of the first word ($p = .0012$) were significant. The trial count was not significant after including a random slope for trial count by participant. The category of problems was

significant (*ps* < .001); a pairwise comparison with Tukey's adjustment revealed that OBJ elicited longer reaction times when choosing answers compared to the other 4 categories.

The model with creativity revealed significant effects of the first word frequency and creativity (*ps* < .001). The trial count was not significant. There were significant differences between categories: OBJ elicited a longer Total Time than other categories (*ps* < .001). Model summaries are available in S2 Appendix.

## Summary of Experiment 2

The analysis of accuracy showed significantly lower accuracy for OBJ compared to other categories in Experiment 2. This is in line with Hayakawa and Keysar's study [22] mentioned in the hypothesis; the use of a foreign language reduces mental imagery of words. The analysis for reaction times revealed that the categorical differences observed in the Total Time arose in the late time frame (in the Solving Time), but not in the early time frame (the Reading Time). This finding is different from the one in Experiment 1 in which the difference in the reaction times between categories had already emerged in the Reading Time. It is assumed that participants in Experiment 1 had already processed the item-related factor that varied from categories when learning the relation of the first domain (A:B). On the other hand, participants in Experiment 2 read problems but might not have processed words deeply enough to learn the relations. For example, simply reading the words *east* and *west* can be accomplished in L1 and L2, but realizing that these two words are in an antonymic relation can be challenging in L2. The qualitative difference between L1 and L2 reading processing is further investigated in the direct comparison between Experiment 1 and 2 and discussed in the general discussion.

Other unique patterns in Experiment 2 were the effects of word frequency. First, with all three analyses on the Total Time, Reading Time, and Solving Time, word frequencies were significant. The significance of word frequencies was observed only in Experiment 2 but not in Experiment 1; word frequencies matter in the time course of word processing and analogical reasoning in L2 but not in L1. Studies have shown that frequency effects are stronger in L2 than in L1 [e.g., 28, 30]. As the effects of frequency tend to be stronger when the L2 vocabulary size is small [54], the proficiency level of the participants in Experiment 2 may have triggered the significant frequency effects. In sum, this study replicated these findings by revealing that people become more sensitive to word frequencies even when solving problems in L2.

Interestingly, the significant effects of word frequencies in Experiment 2 match the discussion on the lack of difference between categories in the Reading time. Frequency plays a significant role when recognizing and naming words [55]. Thus, the significant effects of word frequencies provide partial evidence to support the explanation that participants in Experiment 2 were busy processing words alone and paid less attention to the relations between words in the first domain. This issue with the time course of analogical reasoning in L2 and the frequency effect will be further discussed in the general discussion.

## Direct comparison of languages (L1/L2) with interaction between languages and category

Overall, OBJ items showed a processing disadvantage in accuracy and reaction times; in both L1 and L2; OBJ tended to be read more slowly than the other categories. Further analyses were conducted to investigate the difference between analogical reasoning in L1 and L2 from the perspectives of different types of similarity (perceptual or relational), an interaction between languages (L1/L2) and the category of problems (OBJ, ANT, LTC, MPW, SYN). Model results for accuracy and reaction times are shown in Tables 2–5. The results are visualized in Fig 2.

**Table 2. Mixed-effects model fitted to accuracy of all data with difficulty and an interaction of language and category.**

| | Fixed Effects | | | | | |
|---|---|---|---|---|---|---|
| | Estimate | SE | 95% CI (2.5%) | 95% CI (97.5%) | z | p |
| Intercept | 2.688 | 0.310 | 2.079 | 3.296 | 8.659 | < .001 |
| Category(ANT) | 0.996 | 0.397 | 0.218 | 1.774 | 2.509 | .0121 |
| Category(LTC) | -0.142 | 0.340 | -0.808 | 0.525 | -0.417 | .6770 |
| Category(MPW) | 0.073 | 0.353 | -0.618 | 0.764 | 0.207 | .8358 |
| Category(SYN) | 0.499 | 0.351 | -0.190 | 1.188 | 1.419 | .1558 |
| LANG(ENG) | -1.265 | 0.332 | -1.916 | -0.614 | -3.810 | .0001 |
| Trial count | -0.148 | 0.061 | -0.268 | -0.027 | -2.407 | .0161 |
| Difficulty | -3.075 | 0.131 | -3.332 | -2.818 | -23.446 | < .001 |
| Category(ANT):LANG(ENG) | -0.158 | 0.341 | -0.826 | 0.510 | -0.465 | .6419 |
| Category(LTC):LANG(ENG) | 0.608 | 0.273 | 0.073 | 1.143 | 2.227 | .0259 |
| Category(MPW):LANG(ENG) | 0.483 | 0.290 | -0.086 | 1.051 | 1.665 | .0959 |
| Category(SYN):LANG(ENG) | 0.780 | 0.295 | 0.202 | 1.358 | 2.646 | .0082 |
| | Random Effects | | | | | |
| | Variance | S.D. | | | | |
| Items(Intercept) | 0.717 | 0.847 | | | | |
| Participant(Intercept) | 0.673 | 0.820 | | | | |
| Participant\|Trial count (slope) | 0.119 | 0.346 | | | | |
| Participant\|Difficulty (slope) | 1.376 | 1.173 | | | | |
| | Model fit | | | | | |
| $R^2$ | Marginal | Conditional | AIC | | | |
| | .7 | .789 | 3891.3 | | | |

*Note.* Sub category of problems were abbreviated as ANT(antonym) LTC (location-time context), MPW (material part-whole) and SYN (synonym). Trial count and difficulty were standardized.

**Accuracy.** First, accuracy was analyzed with generalized linear mixed-effects modeling. Fixed effects such as the category of problems, standardized trial count, and standardized rating score of difficulty were included first with random intercepts for participants and items. In addition, an interaction between languages and the category was included. Log-transformed frequencies of words were analyzed in the order that participants encountered them. The final model contained the category, trial count, difficulty, language, and interaction between the category and language as fixed effects. It also contained random intercepts for participants and items and by participant random slopes for trial count and difficulty. The results showed a significant effect of language ($p < .001$), difficulty ($p < .001$), and trial count ($p = .016$). A pairwise comparison with Tukey's adjustment revealed that there was a significant interaction between language and category; the accuracy of OBJ in L1 was significantly higher than that in L2 ($p < .001$), the accuracy of ANT in L1 was significantly higher than that in L2 ($p < .001$), the accuracy of LTC in L1 was significantly higher than that in L2 ($p = .048$), and the accuracy of MPW in L1 was significantly higher than that in L2 ($p = .0228$). The accuracy of SYN in L1 and L2 was not significantly different.

The model with creativity in the place of difficulty revealed a significant effect of creativity and language ($ps < .001$). In this model, trial count was not significant ($p > .05$). Language was significant in all categories: in all categories, solving problems in L1 was significantly more accurate than solving problems in L2 (OBJ ($p < .001$), ANT ($p < .001$), LTC ($p < .001$), MPW ($p < .001$) and SYN ($p = .0103$)).

**Table 3. Mixed-effects model fitted to Total Time of all data with difficulty and an interaction of language and category.**

| Fixed Effects | | | | | | |
|---|---|---|---|---|---|---|
| | Estimate | SE | 95% CI (2.5%) | 95% CI (97.5%) | t | p |
| (Intercept) | 9.000 | 0.055 | 8.892 | 9.108 | 163.795 | < .001 |
| Category(ANT) | -0.259 | 0.041 | -0.340 | -0.178 | -6.274 | < .001 |
| Category(LTC) | -0.172 | 0.042 | -0.254 | -0.090 | -4.106 | < .001 |
| Category(MPW) | -0.200 | 0.040 | -0.279 | -0.121 | -4.970 | < .001 |
| Category(SYN) | -0.216 | 0.040 | -0.295 | -0.138 | -5.399 | < .001 |
| LANG(ENG) | 0.230 | 0.069 | 0.094 | 0.366 | 3.310 | .0013 |
| Trial Count | -0.052 | 0.008 | -0.068 | -0.036 | -6.263 | < .001 |
| Difficulty | 0.252 | 0.022 | 0.208 | 0.295 | 11.359 | < .001 |
| FreqAveragePm_W1_log | -0.014 | 0.006 | -0.025 | -0.003 | -2.570 | .0108 |
| Category(ANT):LANG(ENG) | 0.010 | 0.039 | -0.067 | 0.087 | 0.263 | .7923 |
| Category(LTC):LANG(ENG) | -0.057 | 0.040 | -0.135 | 0.022 | -1.410 | .1585 |
| Category(MPW):LANG(ENG) | -0.048 | 0.038 | -0.123 | 0.027 | -1.258 | .2083 |
| Category(SYN):LANG(ENG) | -0.067 | 0.038 | -0.141 | 0.008 | -1.756 | .0791 |
| Random Effects | | | | | | |
| | Variance | S.D. | | | | |
| Items(Intercept) | 0.009 | 0.095 | | | | |
| Participant(Intercept) | 0.079 | 0.281 | | | | |
| Participant\|Trial count (slope) | 0.004 | 0.061 | | | | |
| Participant\|Difficulty (slope) | 0.032 | 0.178 | | | | |
| Model fit | | | | | | |
| $R^2$ | Marginal | Conditional | AIC | | | |
| | .263 | .401 | 5260.398 | | | |

*Note*. Sub category of problems were abbreviated as ANT(antonym) LTC (location-time context), MPW (material part-whole) and SYN (synonym). Trial count and difficulty were standardized.

Although problems in the four categories such as OBJ, ANT, LTC and MPW were more accurately solved in L1, the accuracies of SYN in L1 and L2 were not significantly different with the model of difficulty as a fixed effect. One interpretation is that in both L1 and L2, selecting the right answer for problems of synonyms (e.g., beginner: novice:: habit: ???) was accomplishable intuitively, without deeply thinking about relations. If so, reaction times between L1 and L2 would be significantly different when the categories were OBJ, ANT, LTC and MPW, but not SYN. To further analyze the language difference in different categories, reaction times were analyzed with an interaction of language and category.

**Total Time (Reading + Solving Time).**   Reaction times were also analyzed with linear mixed-effects modeling in the same manner as the previous analyses but now with the interaction of language and category. First, the Total Time (the sum of the Reading Time and Solving Time) was analyzed. The final model included fixed effects such as category, language, trial count, difficulty, and first-word frequency with an interaction between languages and categories. Random intercepts for participants and items as well as random by participant slopes of the trial count and the difficulty score were included. The results revealed significant effects of difficulty and trial count ($ps < .001$). Language ($p = .0012$) and first-word frequency were also significant ($p = .0107$). A pairwise comparison with Tukey's adjustment revealed that the differences between L1 and L2 with the categories of OBJ ($p < .001$), ANT ($p < .001$), LTC ($p = .0096$), MPW ($p = .0057$) and SYN ($p = .0128$) were significantly different: in all categories, responses were made more quickly in L1 than in L2.

**Table 4. Mixed-effects model fitted to Reading Time of all data with difficulty and an interaction of language and category.**

| Fixed Effects | | | | | | |
|---|---|---|---|---|---|---|
| | Estimate | SE | 95% CI (2.5%) | 95% CI (97.5%) | t | p |
| (Intercept) | 7.859 | 0.080 | 7.702 | 8.015 | 98.485 | < .001 |
| Category(ANT) | -0.113 | 0.032 | -0.176 | -0.050 | -3.497 | < .001 |
| Category(LTC) | -0.073 | 0.033 | -0.138 | -0.008 | -2.196 | .0291 |
| Category(MPW) | -0.080 | 0.031 | -0.141 | -0.018 | -2.545 | .0117 |
| Category(SYN) | -0.086 | 0.031 | -0.147 | -0.024 | -2.742 | .0067 |
| LANG(ENG) | 0.014 | 0.112 | -0.205 | 0.233 | 0.126 | .9000 |
| Trial Count | -0.124 | 0.013 | -0.150 | -0.098 | -9.336 | < .001 |
| Difficulty | 0.061 | 0.010 | 0.042 | 0.079 | 6.348 | < .001 |
| FreqAveragePm_W1_log | -0.016 | 0.004 | -0.025 | -0.007 | -3.497 | < .001 |
| Category(ANT):LANG(ENG) | 0.046 | 0.038 | -0.029 | 0.121 | 1.213 | .2252 |
| Category(LTC):LANG(ENG) | 0.002 | 0.040 | -0.076 | 0.080 | 0.052 | .9585 |
| Category(MPW):LANG(ENG) | 0.036 | 0.038 | -0.039 | 0.110 | 0.941 | .3470 |
| Category(SYN):LANG(ENG) | 0.064 | 0.038 | -0.010 | 0.138 | 1.706 | .0881 |
| Random Effects | | | | | | |
| | Variance | S.D. | | | | |
| Items(Intercept) | 0.003 | 0.056 | | | | |
| Participant(Intercept) | 0.238 | 0.488 | | | | |
| Participant\|Trial count (slope) | 0.012 | 0.111 | | | | |
| Model fit | | | | | | |
| $R^2$ | Marginal | Conditional | AIC | | | |
| | .113 | .202 | 5754.296 | | | |

*Note.* Sub category of problems were abbreviated as ANT(antonym) LTC (location-time context), MPW (material part-whole) and SYN (synonym). Trial count and difficulty were standardized.

A model with creativity in the place of difficulty revealed similar results: there were significant effects of language, trial count, creativity ($ps < .001$) and first word frequency ($p = .0181$). Problems were read and solved significantly faster in L1 than L2 in all categories of OBJ ($p < .001$), ANT ($p < .001$), LTC ($p < .001$), MPW ($p < .001$) and SYN ($p = .0017$). To further investigate when the difference in the time course of analogical reasoning in L1 and L2 emerged, analyses of the Reading Time and Solving Time were conducted.

**Reading Time.** The time that participants spent reading problems A:B::C:??? was analyzed as the Reading Time. The final model included, as fixed effects, the category, language, trial count, difficulty, and first-word frequency with an interaction between languages and category. The model also included random intercepts for participants and items. Adding random slopes for difficulty and trial count by participants in the same model caused a convergence error. A model with a random slope for difficulty and one with trial count were tested with likelihood ratio tests. As a result, the model with a random slope for trial count was chosen as the final model with a significantly smaller AIC. The results revealed significant effects of category, difficulty, trial count, and first-word frequency ($ps < .001$). The effect of language was not significant, and a pairwise comparison with Tukey's adjustment revealed that the differences between L1 and L2 with the categories of OBJ, ANT, LTC, MPW, and SYN were not significantly different ($ps > .05$).

Similar results were observed by a model with creativity in place of difficulty. There were significant effects of trial count, creativity and first-word frequency ($ps < .001$). The effects of language and its interaction between language and category were not significant ($ps > .05$).

**Table 5. Mixed-effects model fitted to Solving Time of all data with difficulty and an interaction of language and category.**

| | Estimate | SE | 95% CI (2.5%) | 95% CI (97.5%) | t | p |
|---|---|---|---|---|---|---|
| Fixed Effects | | | | | | |
| (Intercept) | 8.454 | 0.074 | 8.310 | 8.599 | 114.564 | < .001 |
| Category(ANT) | -0.390 | 0.059 | -0.505 | -0.274 | -6.594 | < .001 |
| Category(LTC) | -0.255 | 0.059 | -0.370 | -0.140 | -4.343 | < .001 |
| Category(MPW) | -0.291 | 0.059 | -0.406 | -0.176 | -4.952 | < .001 |
| Category(SYN) | -0.311 | 0.058 | -0.426 | -0.197 | -5.324 | < .001 |
| LANG(ENG) | 0.270 | 0.096 | 0.081 | 0.459 | 2.804 | .0059 |
| Trial Count | -0.017 | 0.014 | -0.044 | 0.010 | -1.253 | .2141 |
| Difficulty | 0.392 | 0.032 | 0.329 | 0.454 | 12.228 | < .001 |
| Category(ANT):LANG(ENG) | 0.084 | 0.058 | -0.028 | 0.197 | 1.467 | .1425 |
| Category(LTC):LANG(ENG) | -0.043 | 0.058 | -0.156 | 0.070 | -0.742 | .4580 |
| Category(MPW):LANG(ENG) | -0.048 | 0.057 | -0.159 | 0.064 | -0.836 | .4032 |
| Category(SYN):LANG(ENG) | -0.111 | 0.056 | -0.221 | -0.001 | -1.970 | .0488 |
| Random Effects | | | | | | |
| | Variance | S.D. | | | | |
| Items(Intercept) | 0.003 | 0.057 | | | | |
| Participant(Intercept) | 0.239 | 0.488 | | | | |
| Participant\|Trial count (slope) | 0.012 | 0.112 | | | | |
| Participant\|Difficulty(slope) | 0.066 | 0.257 | | | | |
| Model fit | | | | | | |
| $R^2$ | Marginal | Conditional | AIC | | | |
| | .247 | .380 | 10513.130 | | | |

*Note.* Sub category of problems were abbreviated as ANT(antonym) LTC (location-time context), MPW (material part-whole) and SYN (synonym). Trial count and difficulty were standardized.

The analysis on the Reading Time revealed that the difference between L1 and L2 was not statistically significant in both models with difficulty and creativity as fixed effects. This result was surprising because reading in L1 usually tends to be faster than reading in L2, even with very high L2 proficiency [56–58]. Thus, the task was clearly not a simple reading of a word sequence, and the result of the current study was not in line with the ideas of standard language processing. One possible interpretation of the result was that the extent to which participants processed words during the Reading Time was different in L1 and L2. In L1, participants seemed to process words until they sensed the problem-related factors during the Reading Time. For example, they spent more time reading ビスケット：レンズ (biscuit: lens) than reading ヒーロー：腰抜け (hero: coward) in Japanese (L1). In contrast, this was not the same in Experiment 2, as the difference in the Reading Time between categories was not significant. As previously mentioned in the analysis of Experiment 2, no significant difference between categories in addition to the significant effect of word frequency supports the idea that participants in Experiment 2 might process words shallowly when reading problems. In other words, they did not carefully learn the relations of words. In sum, there was not much difference between the Reading Time in L1 and L2 because the language processes were qualitatively different; participants in Experiment 2 processed A:B::C:??? as words at the shallow level and did not pay much attention to the relation of A:B in L2. This would be further described in accordance with the inhibitory effect caused by the use of L2.

**Solving Time.** The duration between the time the participants started to look at multiple choices to the one they chose an answer was analyzed as the Solving Time. The final model

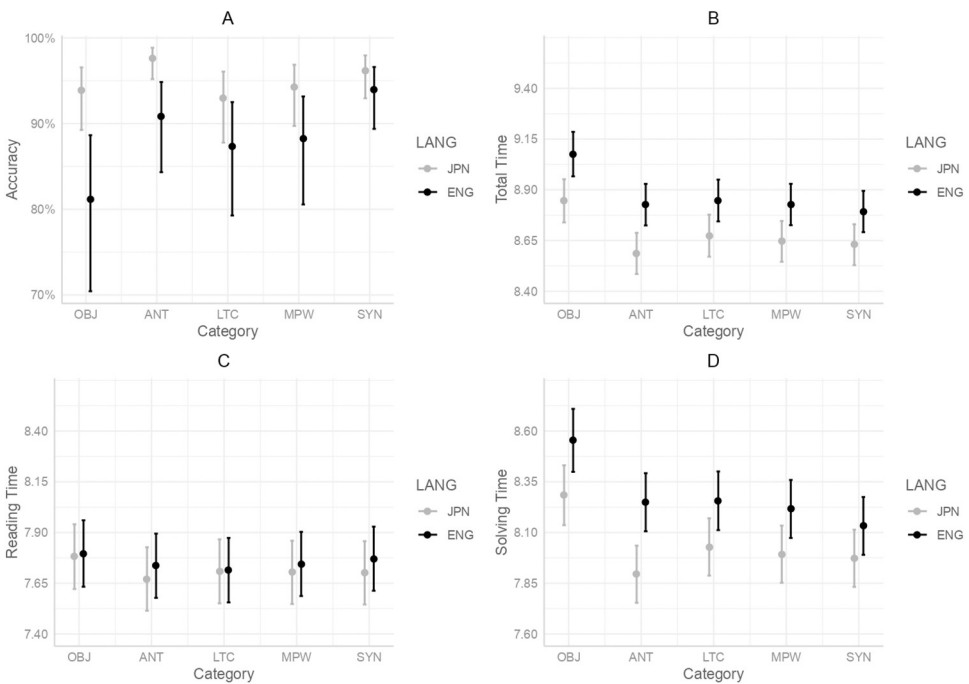

**Fig 2. Interactions between language and category in the mixed-effects models for accuracy and reaction times.** A is a plot of the model for accuracy. B is a plot of the model for Total Time. C is a plot of the model for Reading Time. D is a plot of the model for Solving Time.

included fixed effects such as the category, language, trial count, and difficulty with an interaction between languages and category. Random intercepts of participants and items as well as random slopes of the trial count and the difficulty score by participants were included. The results revealed a significant effect of difficulty ($p < .001$). The effect of trial count was not significant. The effect of language was significant ($p = .0059$), and a pairwise comparison with Tukey's adjustment revealed that the differences between L1 and L2 with the categories of OBJ ($p = .005$), ANT ($p < .001$), LTC ($p = .0127$), and MPW ($p = .0141$) were significantly different. The Solving Times of SYN in L1 and L2 were not significantly different ($p > .05$).

A model with creativity in the place of difficulty revealed similar results: there were significant effects of language ($p = .0049$), trial count ($p = .009$), and creativity ($p < .001$). Problems were read and solved significantly faster in L1 than in L2 in all categories: OBJ ($p = .0043$), ANT ($p < .001$), LTC ($p = .0025$), MPW ($p = .0026$) and SYN ($p = .0333$).

The results of Solving Time with difficulty as a fixed effect revealed that problems of the four categories (OBJ, ANT, LTC, and MPW) elicited less time in L1 than in L2, whereas no such significant difference was observed with the category of SYN. These results were very similar to those of accuracy, as L1 received significantly more accurate responses than L2 in the four categories but not in SYN. The uniqueness of SYN in analogical reasoning in both L1 and L2 is further discussed in the general discussion with reference to previous studies [27, 59–62]. Model summaries for models with creativity are available in S2 Appendix.

## General discussion

We demonstrated that languages (L1/L2) and categories of relations (OBJ, ANT, LTC, MPW and SYN) influence the accuracy and reaction times when solving analogies. In

addition, we also investigated the effects of some linguistic predictors, such as the frequency of words, difficulty, and creativity of problems. In most analyses, difficulty and creativity remained significant for accuracy and reaction times, which is in line with the suggestions from previous studies [2, 27, 34, 35]. Although the previous studies were conducted with items in L1 English, this study found that those predictors contributed to analogical reasoning in a different language (L1 Japanese) and with second-language users (L2 English). Analyses revealed that models with difficulty had a larger R-squared than those with creativity. We discuss other findings in the current study by answering three research questions stated in the Introduction.

## Does the language difference (L1/L2) influence the reaction times and accuracy of analogical reasoning?

A direct comparison between the experimental results of L1 and L2 revealed that solving analogical problems in L1 was more accurate and faster than in L2. The results were in line with some previous studies that suggested the inhibitory effect of foreign (unfamiliar) language use while reasoning [14]. The current study offers an explanation for why and how language unfamiliarity negatively influences the processing of analogies. In particular, our findings highlight the possibility that the use of less familiar language might cause the inefficacy of learning relations at the very beginning of the problem-solving process. Learning A:B relations is fundamental to effectively stimulate C:D. The eye-tracking experiment in Vendetti et al. [63] revealed that the primary focus of A:B relation is fundamental for accurate responses (see also [64] for the review of processing models of analogical reasoning). In our study, all participants were adults, but the difference was the languages (L1/L2) they used when reading and solving analogies. As mentioned earlier, the results diverged between the experiments: differences in categories were significant in the analysis of the Reading Time in Experiment 1 but not in Experiment 2. Thus, people process relations of A:B when reading problems in L1 but not in L2. A direct comparison of those results also revealed a nonsignificant difference between the reading times of L1 and L2. This result was contrary to some previous studies that assumed reading in L1 usually tends to be faster than reading in L2 [56–58]. However, with the different results of the Reading Time in Experiments 1 and 2, the reading behavior in L1 and L2 might have been qualitatively different: although reading L1 entailed deeper processing of A:B relations, this was not the case in L2. Moreover, the significant effect of frequencies in Experiment 2 also explains why processing in L2 would be shallower than in L1, as frequency plays a significant role in word recognition [55]. Taken together, the results of the present study explain that one of the aspects of cognitive load in L2 reasoning is most likely related to the inefficiency of word recognition, and that makes participants focus on reading words but not processing relations of words.

## To what extent do perceptual and relational similarities influence the reaction times (RTs) and accuracy of verbal analogical reasoning?

Although differences between OBJ and other categories varied across analyses, a tentative conclusion can be made, as OBJ requires longer response times according to the results of Experiments 1 and 2. A potential explanation is that accessing visual information with verbal input is more cognitively demanding than accessing semantic information. As mentioned in the Introduction, a general framework introduced by Huettig et al. [26] states that linguistic information activates phonological and semantic structures first and then their associated conceptual visual representations such as colors and shapes. Solving OBJ requires both activation of semantic structures and visualization of the conceptual shape of words, whereas solving other

categories (e.g., ANT, LTC, MPW, and SYN) can be settled while activating semantic structures. For this reason, it is reasonable that OBJ had a longer response time in both Experiments 1 and 2. Although OBJ may have a longer response time due to the additional cognitive processing of words in long-term memory, according to their explanation, it would be possible to access visual information of words as long as enough processing time is given. As expected, accuracies between OBJ and other categories in Experiment 1 were not significantly different. However, in Experiment 2, OBJ received more inaccurate responses than SYN even when difficulty was included as a fixed effect. Therefore, simulating shape similarity in mental images is more challenging when having L2 input than L1 input. These results are in line with Hayakawa and Keysar's study [22]. Their results revealed that it was more difficult for participants to find similar shaped objects than categorically similar objects when stimuli were L2 verbal inputs. Moreover, although they found that responses in L1 were more accurate than in L2 in both the shape task and category task, the language effect was larger in the shape task. Their results imply that visualization in L2 imposes a greater cognitive load than in L1. In sum, with verbal stimuli, accessing the perceptual similarity of word concepts is more challenging than finding relational similarity. In addition, using a foreign language may further reduce the efficacy of the visualization of words compared to using a native language.

## Is there a significant interaction between the similarities and languages?

In addition to the finding of the characteristics of perceptual and relational similarity in analogical reasoning in L1 and L2, analyses of the interaction between languages and categories revealed another interesting feature of the item category. In the four categories of OBJ, ANT, LTC and MPW, L1 elicited better accuracies and faster Solving Times than L2, whereas such a difference was not observed with SYN. This result was partially in line with Jones et al. [27], as they observed higher accuracies for categorical analogies (e.g., hammer: tool) than for compositional and causal analogies (e.g., allergy: sneeze). Although they already mentioned that spontaneous activation of category-related relations facilitated analogical reasoning in L1, our results confirmed that the tendency can also be seen in analogical reasoning in L2.

The similar phenomenon has been reported in some studies on the mental lexicon. In the studies of the mental lexicon, "superordinates (dog-animal), subordinates (dog-terrier), and synonyms (dog-canine)" are categorized as paradigmatic information [62, p.43]. In some word association tasks, such paradigmatic information is more likely to be produced than other information such as syntagmatic information (e.g., "house-big," and "tall-giraffe") [60, p. 664]. Nissen and Henriksen [61] reported that nouns elicit paradigmatic information more than other word classes, such as verbs and adjectives. The problems used in this study consisted of nouns. This might have caused facilitation for SYN relations due to the lexical information that can be readily evocative with nouns. For this reason, why SYN was easier might be due to parts of speech in this study. Investigation of the effects of parts of speech in analogical reasoning is beyond our research scope, but it would be worth exploring in future studies.

## Implications

Some implications for future study must be addressed before conclusion. One of them is the high correlation between subjective difficulty and creativity. As mentioned earlier, creativity refers to the semantic distance between relations of the first domain (A:B) and the second domain (C:D), and the distance can be usually obtained from a database. Such a database does not exist in L2, so rating tasks were conducted to collect subjective difficulty and creativity. To avoid such a correlation, rating tasks were conducted separately with an adequate experimental design by randomizing the order of item presentation. Even so, the result revealed a high

correlation between the ratios ($r$ = .95, $p < $ .001). One possible explanation is that participants interpreted those ratio in the same way (e.g., *Rate how difficult the problem was* and *Rate how surprising the C:D pair was, given the A:B pair*). As a result, those two ratios might have been basically reflecting the same aspect of the analogical reasoning problems. Another explanation is that difficulty and creativity are two different aspects but are posited in a hierarchical manner in which difficulty is considered the superordinate of creativity. Therefore, participants might have found a problem difficult, and that sense of difficulty made them think the problem was creative. Studies on related factors on analogical reasoning problems have started only recently and the relations between these factors have not been fully investigated yet. Multivariate analysis with those factors to investigate the relations between them is encouraged for future studies.

Although the results of this study suggested that relational similarity is more accessible than perceptual similarity when using verbal stimuli, it is important to note that this finding may be influenced by the unequal number of items across similarity conditions. The study included 100 problems based on five different relational structures: SYN (e.g., beginner: novice), ANT (e.g., hero: coward), MPW (e.g., skeleton: bone), LTC (e.g., knight: castle), and OBJ (e.g., tomato: ball). Each category consisted of 20 problem sets. Four categories (synonyms, antonyms, material part-whole, and location-time context) were created based on relational similarity, while object matches were created based on perceptual similarity. The number of items for relational and perceptual similarity was not balanced, with 80 problems falling under the relational similarity category and 20 problems under perceptual similarity category. Future studies are encouraged to prepare the same number of items for each similarity condition. Having an equal number of items for each condition (similarity) is not straightforward because it is challenging to determine which category (ANT, SYN, LTC, or MPW) represents the typical relational structure. One suggestion that this study would like to offer is that using a certain category, such as SYN, can induce potential facilitatory effects compared to other categories. As already mentioned, coming up with synonym word (e.g., woman) by seeing a target word (e.g., lady) can be achievable without a contribution of analogical mapping from the first domain (e.g., delight: happiness). If problems can be solved without focusing on the first domain, the observed process would likely only involve a simple lexical association, or in other words, more likely the "System 1" processing [65, p.658]. This contrasts with analogical reasoning, which entails rule-based information mapping from the first to the second domains, and can be regarded as the "System 2" processing [65, p.658]. To consider the potential processing differences among categories, researchers are encouraged to pay attention to how much focus on the first domain would contribute to the accuracy of responses when preparing problems.

Furthermore, it is possible that analogy problems with perceptual similarity may not represent the standard form of analogies. In other words, participants may have found that literal shape similarity is not a typical form of analogy. To address this issue, it may be more appropriate to employ problem-solving tasks with stories or longer texts. For instance, Wakebe et al. [14] used The Ray Problem and asked readers to find a solution to destroy a tumor using X-rays without causing harm to the healthy tissue surrounding it. There are structurally analogous stories such as the firefighter's problem, the fortress problem, and the lightbulb problem introduced in the previous studies [4, 66, 67]. Although the specific objectives may differ across stories, the underlying solution to these problems remains structurally analogous: dividing power to solve the problems without damaging the objects. While the facilitatory effect of perceptual similarity was not supported in this study, it is still possible that perceptually similar objects, such as fortresses and houses, may still enhance the analogical mapping between domains even when verbal inputs are provided.

Finally, as a reviewer pointed out, theories focusing primarily on working memory may better explain the patterns observed in this study. Working memory is crucial in analogical reasoning because it requires storing and transferring information, such as similarities, across domains [68]. We did not address this issue because our methodology—presenting A:B::C:D alongside response options—did not control memory retention during reasoning. Nevertheless, a thorough examination of the effects of working memory requires a careful approach in experimental settings. Future studies employing time-sensitive methods, such as tracking eye fixations or PC mouse movements in a well-controlled environment, would offer valuable insights into the relationship between working memory and L2 analogical reasoning.

## Conclusion

Like a metaphor as an example in the beginning, verbal analogical reasoning plays a significant role in acquiring new words. Previous research found a cognitive load due to L2 use in reasoning; however, the details of processing difficulties in L2 reasoning were not systematically investigated. Our study explored the impact of language and categories of relations on the accuracy and time course of analogical reasoning. Consistent with prior research, we found that using a foreign language can negatively influence analogical reasoning across most categories. In addition, our results on the Reading Time indicated that people can process relations between words in L1 without any specific instruction. On the other hand, processing relations between words in L2 would be much more challenging possibly due to the cognitive demand related to lexical processing. These results imply that language plays a crucial role in determining the speed, accuracy, and manner with which individuals can perform analogical reasoning. Taking them into account, giving L2 learners efficient time to process words and explicitly instructing them to understand the relations of words when making analogies is expected in an educational setting. Furthermore, our results indicated that the category of relations also has an influence on analogical reasoning. Notably, relations of OBJ were found to be more difficult while SYN were easier for both L1 and L2 analogical reasoning, which was in line with studies on analogical reasoning [27] and mental lexicon [60]. This highlights the potential significance of lexical predictors in the study of analogical reasoning. As such, our findings shed light on the importance of investigating analogical reasoning in L1 and L2 in relation to predictors such as problem categories and word frequency.

## Supporting information

**S1 Appendix. This document includes a description of the word association task on page 11.**
(DOCX)

**S2 Appendix. This document includes plots and tables of the results of the modeling analyses for Experiments 1 and 2.** It also contains tables from the final models with the effect of creativity as a fixed effect.
(DOCX)

## Author Contributions

**Conceptualization:** Miki Ikuta.

**Data curation:** Miki Ikuta.

**Formal analysis:** Miki Ikuta.

**Funding acquisition:** Miki Ikuta.

**Investigation:** Miki Ikuta.

**Methodology:** Miki Ikuta, Koji Miwa.

**Project administration:** Miki Ikuta.

**Resources:** Miki Ikuta.

**Software:** Miki Ikuta.

**Supervision:** Koji Miwa.

**Validation:** Miki Ikuta.

**Visualization:** Miki Ikuta.

**Writing – original draft:** Miki Ikuta.

**Writing – review & editing:** Miki Ikuta, Koji Miwa.

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
