## [Decision Letter · Decision Letter 0]

18 Sep 2024

PONE-D-24-30659Analogical reasoning in first and second languagesPLOS ONE

Dear Dr. Ikuta,

Thank you for submitting your manuscript to PLOS ONE. After careful consideration, we feel that it has merit but does not fully meet PLOS ONE’s publication criteria as it currently stands. Therefore, we invite you to submit a revised version of the manuscript that addresses the points raised during the review process.

Please take particular care to the comments raised by the reviewers, in particular:

- The justification, objectives and motivations for this research work should be articulated more.

- The methodology and its background would require further, deeper investigation. The links with the related literature should be more explored as well, highlighting the novelty of this paper. Be sure to cite the latest and more related references in the literature.

- Explain better and motivate the employed dataset and features, in particular the choice of the used data sample size. 

- The reported computational analysis and discussion show to be vague and not robust enough, they should be articulated more.

- The manuscript structure needs also a substantial rearrangement, as outlined by the reviewers. Overall readability and presentation should be improved as well. 

Unfortunately the manuscript fails to meet the PLOS ONE publication criteria at this stage. 

We look forward to receiving your revised manuscript.

Kind regards,

Sergio Consoli

Academic Editor

PLOS ONE

2. Thank you for stating the following financial disclosure: [Kaken Grant-in-Aid for Japan Society of Promotion of Science Fellows (21J15484)]. Please state what role the funders took in the study. If the funders had no role, please state: "The funders had no role in study design, data collection and analysis, decision to publish, or preparation of the manuscript." If this statement is not correct you must amend it as needed. Please include this amended Role of Funder statement in your cover letter; we will change the online submission form on your behalf.

3. We note that you have indicated that there are restrictions to data sharing for this study. For studies involving human research participant data or other sensitive data, we encourage authors to share de-identified or anonymized data. However, when data cannot be publicly shared for ethical reasons, we allow authors to make their data sets available upon request. For information on unacceptable data access restrictions, please see http://journals.plos.org/plosone/s/data-availability#loc-unacceptable-data-access-restrictions. Before we proceed with your manuscript, please address the following prompts: a) If there are ethical or legal restrictions on sharing a de-identified data set, please explain them in detail (e.g., data contain potentially identifying or sensitive patient information, data are owned by a third-party organization, etc.) and who has imposed them (e.g., a Research Ethics Committee or Institutional Review Board, etc.). Please also provide contact information for a data access committee, ethics committee, or other institutional body to which data requests may be sent. b) If there are no restrictions, please upload the minimal anonymized data set necessary to replicate your study findings to a stable, public repository and provide us with the relevant URLs, DOIs, or accession numbers. Please see http://www.bmj.com/content/340/bmj.c181.long for guidelines on how to de-identify and prepare clinical data for publication. For a list of recommended repositories, please see https://journals.plos.org/plosone/s/recommended-repositories. You also have the option of uploading the data as Supporting Information files, but we would recommend depositing data directly to a data repository if possible. Please update your Data Availability statement in the submission form accordingly.

Additional Editor Comments (if provided):

Reviewers' comments:

Reviewer's Responses to Questions

**Comments to the Author**

1. Is the manuscript technically sound, and do the data support the conclusions?

Reviewer #1: Yes

Reviewer #2: Yes

Reviewer #3: Partly

Reviewer #4: Yes

2. Has the statistical analysis been performed appropriately and rigorously? 

Reviewer #1: Yes

Reviewer #2: Yes

Reviewer #3: Yes

Reviewer #4: Yes

3. Have the authors made all data underlying the findings in their manuscript fully available?

Reviewer #1: Yes

Reviewer #2: Yes

Reviewer #3: No

Reviewer #4: Yes

4. Is the manuscript presented in an intelligible fashion and written in standard English?

Reviewer #1: Yes

Reviewer #2: Yes

Reviewer #3: Yes

Reviewer #4: Yes

5. Review Comments to the Author

Reviewer #1: Great manuscript but incredibly lengthy. Would recommend this be broken up into at least two manuscripts, or at the very least have clearly defined sections. For example, the methods include results and results include methods and discussion items. This makes it difficult to follow the content of this analysis. As an aside, there should be continuous numbering throughout.

Introduction - would make this significantly more succinct. Line 6-7: "Foot tapping". This can be omitted, as it does not appear to contribute anything to the introduction.

Methods - would remove all results (e.g lines 15-19). Statistical methods should be outlined in the methods section, not in the results section (e.g., lines 18-24, 1-17)).

Results - only report results, not items for the discussion (e.g.,lines 22-23 "this issue will be further discussed in the general discussion, recalling Hayakawa and Keysar’s study [22]). This is not necessary.

Reviewer #2: This is a timely and useful study that provides important predictors for analogical reasoning in bilinguals. It focuses on how analogies are processed in a second language and makes use of linguistic predictors like word frequencies, difficulty and creativity problems to show how they contribute to analogical reasoning. The study portrays how these predictors are useful in influencing performance of individuals’ analogical reasoning in first and second languages. The findings show among others that using a less familiar language hinders efficacy of analogical reasoning. Similarly, first language elicited higher accuracies and faster response time than the second language. This expands our understanding of how analogical reasoning is less detailed or effective when a second less familiar language is introduced into the equation. While this study is useful in many respects and to a large extent well executed, it can be improved to make it more robust. The following are observations and suggestions for improvement:

1. There is an over use of examples in the background of the write up: the mitochondria analogy and the ray problem for instance are useful but have been repeated rendering them redundant without advancing the argument. It will be better to introduce them once and refer to them without explaining them in detail each time.

2. Some terms have been introduced without sufficient clarification. Examples include ‘relational similarity’ and ‘perceptual similarity’. These are important arguments that necessitate the study, yet they are only briefly explained. A detailed explanation is needed. Also, phrases like ‘analogical transfer is language-free’ needs to be in context. Without this contextualization it is not clear as proficiency differences can also play a role.

3. The study makes reference to previous studies but does not really critically engage with them. This can be resolved by looking at their findings, contradictions and examining potential reasons for these outcomes in relation to the present study.

4. It will be beneficial to explain why reaction times are important to the study as the research questions place strong emphasis on this. For instance, how does reaction time contribute to understanding the cognitive processes behind analogical reasoning and why this is an important metric.

5. In the hypothesis, there is a focus on differences in accuracy and reaction time between L1 an L2 but this lacks specificity. For instance, there is the hypothesis that ‘lower accuracy is expected for the L2 condition’. This could be made more precise by stating the expected differences between specific relational times or levels of proficiency.

6. Cognitive load associated with L2 processing is mentioned but not discussed in depth. It appears that cognitive load is an important factor in understanding differences in L1 and L2 performance and so a detailed discussion of how it influences analogical reasoning could strengthen the argument.

7. Take note of in-text citations. For instance, some citations have page numbers for direct quotes but others do not. There should be consistency.

8. In terms of the methodology, how were variables such as participants’ environment and potential distractions controlled for during tasks. These could affect time and accuracy.

9. The study emphasizes the importance of word frequency and what it does for L2 analogical reasoning. However, it missed explaining how specific frequency thresholds like high versus low frequency were determined. If this is available, it is less clear. This makes it quite difficult to see consistency and relevance of frequency selections across tasks.

10. Inconsistent definition of ‘difficulty’ and ‘creativity’ makes it less clear how each term is unique in its own way in terms of cognitive load. The overlap between the two (difficulty =problem solving and creativity=semantic distance) is not detailed. How was it ensured that participants see them as separate constructs during rating tasks?

11. Cultural considerations have been shown to some extent but not adequately accounted for in L1 and L2 processing. The example of ‘filament’ used from previous studies may be unfamiliar due to cultural differences and not linguistic differences. This makes the study’s assumption about word frequency as a purely linguistic factor problematic.

12. The methodology does not address issues of cognitive load if any, like attentional demands and anxiety in processing analogies in L2.

13. It is not clear from the manuscript how the 100 analogical problems were ordered or presented. If this is present in the study, it needs to be clear in the methodology.

Overall, the authors need to improve the manuscript by streamlining the background of the study and rearranging the flow of information in the manuscript for consistency. They also need to critically engage the literature and clarify the hypothesis. It should be clear in the methodology section how control over word selection, clarity in defining predictors, addressing cultural factors and cognitive load was done. This information should not be nested in the analysis but stand out in the methodology. Once these issues are addressed, the study will be firmed and improved.

Reviewer #3: The manuscript attempts to make a contribution to the existing literature on analogical problem-solving among bilinguals. There is indeed a growing body of research in this area. In this manuscript, the authors conducted two experiments among bilinguals with the first experiment requiring them to complete behavioural tasks in their native language (Japanese) while the second experiment presented a series of tasks for them to complete in their second language (English). The manuscript was largely well written except that I have some few concerns which I believe when addressed will substantially improve its suitability for publication. These issues are catalogued below;

1. Clearly numerous studies have been conducted on this subject. So, it is not clear what specific knowledge will be derived from this study. The study failed to highlight the novelty of the research conducted. What specific knowledge gap does this study seek to address? This should be emphasized in the early part of the introduction.

2. The study failed to include any theory in the literature review.

3. On methodology, in both experiments, participants completed the tasks online. It is not clear at what time, location and devices (PCs, mobile phones, tablets?) were used to complete the tasks. Since this is an experiment, what control measures put in place to eliminate potential extraneous variables that could impact the results?

4. Moreover, the sample size is very small. What is the justification for the use of this small sample size? I recommend a larger sample size.

5. The presentation of the results in the text largely does not follow the APA format. I see that it is mostly the p-values that are stated. This should be more than that.

Reviewer #4: This paper provides an insightful exploration of analogical reasoning and the differences between processing analogies in first (L1) and second (L2) languages. Although I come from a background in mathematics and computational social behavior, I found the manuscript accessible and well-written, allowing me to easily follow the arguments and conclusions. The authors clearly cover relevant prior literature, provide strong motivation for their study, and pose interesting research questions.

The paper focuses on whether language differences (L1 vs. L2) influence reaction times and accuracy in analogical reasoning. Additionally, the authors examine the extent to which perceptual and relational similarities affect these two metrics and whether there is a significant interaction between similarity types and languages. The paper approaches these questions by conducting two experiments with bilingual speakers of Japanese (L1) and English (L2).

The results show that analogical reasoning in L1 (Japanese) leads to both faster response times and higher accuracies compared to L2 (English). This suggests that participants are more efficient in their native language when performing cognitive tasks that require complex reasoning. The authors also find that perceptual similarity (e.g., object shape) is particularly challenging when processed in L2, likely due to the increased cognitive load involved in mentally simulating visual features in a foreign language.

As I already mentioned, I am not in this field, so I apologize if my suggestions and questions are not fully relevant. I have some questions and suggestions for the authors:

1. The sample sizes of 42 and 41 seem relatively small. Was a power analysis conducted to justify the number of participants?

2. Some sentences in the abstract, such as “In this experiment, participants also rated the difficulty and creativity of problems” and “A significant interaction was found between languages (L1/L2) and the category of problems,” state facts but do not inform the reader of the actual results. For example, the fact that perceptual similarity is particularly challenging when processed in L2 is not mentioned in the abstract. I believe the current abstract does not do justice to the results, and I would recommend the authors revisit it.

3. The authors begin the abstract by stating, “Analogical reasoning plays a significant role in language learning.” Could the authors elaborate on the implications of their findings, such as how they apply to language learning, for example in educational settings or in designing better tools for bilingual individuals? This could be addressed in the conclusion.

4. One disadvantage of the paper is its length. While the authors have organized it well with sections and subsections that clearly convey specific topics and summarize the results as they go, it remains a lengthy text without much visual aid. I suggest adding a conceptual figure summarizing the experiments. For instance, a visual could show that there are two language groups, the A:B:C format (number of questions), the different categories being tested, and the fact that you are measuring both accuracy and response time. Please note that this is simply a recommendation.

6. PLOS authors have the option to publish the peer review history of their article (what does this mean?). If published, this will include your full peer review and any attached files.

Reviewer #1: No

Reviewer #2: No

Reviewer #3: No

Reviewer #4: No

---

## [Author Response · Author response to Decision Letter 0]

6 Nov 2024

We uploaded responses to reviewers as a separate file labeled 'Response to Reviewers'.

---

## [Decision Letter · Decision Letter 1]

15 Jan 2025

Analogical reasoning in first and second languages

PONE-D-24-30659R1

Dear Dr. Ikuta,

We’re pleased to inform you that your manuscript has been judged scientifically suitable for publication and will be formally accepted for publication once it meets all outstanding technical requirements.

There were some remaining minor points highlighted by two reviewers which I invite you to address in the final version of your manuscript.

Kind regards,

Sergio Consoli

Academic Editor

PLOS ONE

Additional Editor Comments (optional):

I'm suggesting acceptance at this stage because I judge the paper suitable for publication without requiring an additional review cycle of the manuscript. However, there were some remaining minor points highlighted by two reviewers which I invite you to address in the final version of your manuscript.

Reviewers' comments:

Reviewer's Responses to Questions

**Comments to the Author**

1. If the authors have adequately addressed your comments raised in a previous round of review and you feel that this manuscript is now acceptable for publication, you may indicate that here to bypass the “Comments to the Author” section, enter your conflict of interest statement in the “Confidential to Editor” section, and submit your "Accept" recommendation.

Reviewer #1: All comments have been addressed

Reviewer #2: (No Response)

Reviewer #3: (No Response)

2. Is the manuscript technically sound, and do the data support the conclusions?

Reviewer #1: Yes

Reviewer #2: Yes

Reviewer #3: Yes

3. Has the statistical analysis been performed appropriately and rigorously? 

Reviewer #1: Yes

Reviewer #2: Yes

Reviewer #3: Yes

4. Have the authors made all data underlying the findings in their manuscript fully available?

Reviewer #1: Yes

Reviewer #2: Yes

Reviewer #3: No

5. Is the manuscript presented in an intelligible fashion and written in standard English?

Reviewer #1: Yes

Reviewer #2: Yes

Reviewer #3: Yes

6. Review Comments to the Author

Reviewer #1: (No Response)

Reviewer #2: The revised manuscript to some extent effectively introduces analogical reasoning as a cognitive process and explains its application in various domains, such as education, psychology, and artificial intelligence. The examples, like the mitochondria metaphor, clearly illustrate the concept.

Specific gaps in the literature have also been better clarified, particularly regarding the underexplored mechanisms of analogical reasoning in second-language (L2) contexts. This sets the stage well for study.

By referencing seminal studies (e.g., Francis, Wakebe et al.), the study establishes a strong theoretical and empirical foundation. This lends credibility to the research questions.

The research questions are specific, addressing reaction times, accuracy, types of similarity (perceptual and relational), and interactions between language and similarity. These questions align well with the identified research gaps.

The introductory part also integrates findings from related domains, such as lexical processing and visual-semantic interaction, to provide a broader context for the study.

However,

1. Some points, such as the role of perceptual and relational similarities, are reiterated across sections. This could be streamlined to enhance readability and make the manuscript concise.

2. While the research questions are clear, the hypotheses still need to be articulated more explicitly, particularly in linking prior findings to the expected outcomes in the present study.

3. Also, the shifts from general analogical reasoning to bilingual processing and then to perceptual/relational similarity are abrupt. Improved transitions would help maintain coherence.

4. It will be useful to group related discussions under same headings, such as "Analogical Reasoning and Language" and "Role of Similarity in Cognitive Processes," to improve the flow.

5. Use linking sentences to connect sections, such as explaining how findings in lexical processing inform the study of analogical reasoning in L2 contexts.

The authors still need to firm up these observations made for the manuscript to be robust.

Reviewer #3: 1. The point on theoretical review was not addressed. I think that there are theories in cognitive science that can best explain reasoning within this context. The success of tasks of this nature can be explained by working memory and other Executive Functions and so I expect that the authors would have discussed this point and support why we may or not find differences in the processing by the relevant theory(ies).

2. On the methodology regarding the control measures used in the experiment, the answer is not satisfactory. However, this can be stated in the discussion as a limitation of the study.

7. PLOS authors have the option to publish the peer review history of their article (what does this mean?). If published, this will include your full peer review and any attached files.

Reviewer #1: No

Reviewer #2: No

Reviewer #3: No

---

## [Editor Report · Acceptance letter]

1 Feb 2025

PONE-D-24-30659R1 

PLOS ONE

Dear Dr. Ikuta, 

I'm pleased to inform you that your manuscript has been deemed suitable for publication in PLOS ONE. Congratulations! Your manuscript is now being handed over to our production team.

Kind regards, 

on behalf of

Dr. Sergio Consoli 

Academic Editor

PLOS ONE